# The Regulatory Roles of *Ezh2* in Response to Lipopolysaccharide (LPS) in Macrophages and Mice with Conditional *Ezh2* Deletion with LysM-Cre System

**DOI:** 10.3390/ijms24065363

**Published:** 2023-03-10

**Authors:** Areerat Kunanopparat, Asada Leelahavanichkul, Peerapat Visitchanakun, Patipark Kueanjinda, Pornpimol Phuengmaung, Kritsanawan Sae-khow, Atsadang Boonmee, Salisa Benjaskulluecha, Tanapat Palaga, Nattiya Hirankarn

**Affiliations:** 1Center of Excellence in Immunology and Immune-Mediated Diseases, Chulalongkorn University, Bangkok 10330, Thailand; 2Department of Microbiology, Faculty of Medicine, Chulalongkorn University, Bangkok 10330, Thailand; 3Center of Excellence in Translational Research in Inflammation and Immunology (CETRII), Faculty of Medicine, Chulalongkorn University, Bangkok 10330, Thailand; 4Division of Nephrology, Department of Medicine, Faculty of Medicine, Chulalongkorn University, Bangkok 10330, Thailand; 5Department of Microbiology, Faculty of Science, Chulalongkorn University, Bangkok 10330, Thailand

**Keywords:** sepsis, lipopolysaccharide, macrophages, epigenetics, *Ezh2*

## Abstract

The responses of macrophages to lipopolysaccharide (LPS) might determine the direction of clinical manifestations of sepsis, which is the immune response against severe infection. Meanwhile, the enhancer of zeste homologue 2 (*Ezh2*), a histone lysine methyltransferase of epigenetic regulation, might interfere with LPS response. Transcriptomic analysis on LPS-activated wild-type macrophages demonstrated an alteration of several epigenetic enzymes. Although the *Ezh2*-silencing macrophages (RAW264.7), using small interfering RNA (siRNA), indicated a non-different response to the control cells after a single LPS stimulation, the *Ezh2*-reducing cells demonstrated a less severe LPS tolerance, after two LPS stimulations, as determined by the higher supernatant TNF-α. With a single LPS stimulation, *Ezh2* null (*Ezh2*^flox/flox^; LysM-Cre^cre/−^) macrophages demonstrated lower supernatant TNF-α than *Ezh2* control (*Ezh2*^fl/fl^; LysM-Cre*^−/−^*), perhaps due to an upregulation of *Socs3*, which is a suppressor of cytokine signaling 3, due to the loss of the *Ezh2* gene. In LPS tolerance, *Ezh2* null macrophages indicated higher supernatant TNF-α and IL-6 than the control, supporting an impact of the loss of the *Ezh2* inhibitory gene. In parallel, *Ezh2* null mice demonstrated lower serum TNF-α and IL-6 than the control mice after an LPS injection, indicating a less severe LPS-induced hyper-inflammation in *Ezh2* null mice. On the other hand, there were similar serum cytokines after LPS tolerance and the non-reduction of serum cytokines after the second dose of LPS, indicating less severe LPS tolerance in *Ezh2* null mice compared with control mice. In conclusion, an absence of *Ezh2* in macrophages resulted in less severe LPS-induced inflammation, as indicated by low serum cytokines, with less severe LPS tolerance, as demonstrated by higher cytokine production, partly through the upregulated *Socs3*.

## 1. Introduction

Characteristics of sepsis (a potentially life-threatening condition in response to severe infection) are surprisingly similar, regardless of the organismal causes of the infection, and sepsis from bacteria, viruses, fungi, or parasites can be fatal [1,2,3]. Sepsis-induced immune dysfunction is roughly divided into hyperinflammation and immune exhaustion (immune paralysis) [4,5], with some differences in the characteristics. While sepsis-induced hyperinflammation possibly leads to the dysfunction of several organs from hypercytokinemia, immune exhaustion causes secondary infection from inadequate inflammation to control the organism, resulting in another episode of severe infection-induced sepsis [6]. As such, direction of the clinical manifestations in sepsis in the individual patient may be determined by balance of the immune responses, which appear to occur simultaneously in the same patients [7]. A suitable immune regulation, such as an anti-inflammatory treatment during sepsis-hyperinflammation and an escalation of immune responsiveness during immune exhaustion, may be helpful [8,9,10,11,12,13,14,15]. Nevertheless, sepsis-induced immune exhaustion with secondary infection may become more common in patients with sepsis due to the improved supportive care for the maintenance of patients during hyper-inflammatory sepsis, which decreases mortality, especially in the early phase of sepsis [16]. Although several factors are mentioned as the cause of sepsis-induced immune exhaustion, including apoptotic death of several immune cells, myeloid-derived suppressor cells, and increased regulatory T cells, data on lipopolysaccharide (LPS) tolerance, which is defined as the decreased responses following secondary or prolonged LPS stimulation [17,18,19,20], are relatively more rare compared with other mechanisms.

During sepsis, the translocation of LPS, which is a major molecule of Gram-negative bacteria (the most abundant gut organism), from the intestine into the blood circulation, referred to as “leaky gut”, is a common cause of endotoxemia [21,22,23]. Meanwhile, an adaptation to the prolonged LPS stimulations in sepsis may initiate LPS tolerance [24,25]. Then, LPS tolerance might, at least in part, be important in sepsis-induced immune exhaustion. Interestingly, underlying mechanisms of LPS tolerance, especially in monocytes or macrophages, are still unclear, and possibly consist of epigenetic modifications, chromatin remodeling, and interferences in cell energy status [26,27,28]. Among these, epigenetic alterations in LPS stimulation and LPS tolerance have been extensively studied [29,30]. Epigenetics, phenotypic alterations without changes in the DNA sequence, are the processes for the switch on and off of DNA transcription through DNA methylation (adding the methyl group into the DNA), histone modification, and noncoding RNA action (microRNA) [2]. DNA methylation and histone modification are processed through three groups of enzymes, including (i) the writers’ epigenetic regulation (methylation, acetylation, phosphorylation, ubiquitination), (ii) the epigenetic erasers for removing the modifications, and (iii) the epigenetic readers for the binding to different covalent modifications by the writers to mediate physiological outcomes [3]. Among all, the methylation at histone 3 lysine 27 (H3K27) is one of the most common histone changes during several cell activities, including activation by LPS [31].

After LPS stimulation, the attachment of methyl groups at lysine 27 on histone 3 (H3K27) by histone demethylase is induced in macrophages by the Polycomb repressor complex group 2 (PCR2), which belongs to the Chromobox family proteins that mediate gene silencing (a repressor of the transcriptional activity). Then, PCR2 might be associated with (i) the control of overwhelming production of cytokines and other proteins after LPS stimulation as counter anti-inflammation and/or (ii) the too-low productivity of cytokines and other molecules in LPS tolerance [32,33]. Thus, the methylation of histone (H3K27) by the PCR2 complex, which consists of several subunits, including Ezh2 (histone-lysine N-methyltransferase-2 or Enhancer of Zeste Homolog), might reduce cytokine production through the switch-off of DNA transcription [34,35]. Because Ezh is an important catalytic subunit for the methylation process, the over-expression of Ezh enhances PCR2 function, leading to anti-inflammatory properties [36,37] and the blockage of Ezh enhances pro-inflammatory responses [38]. Interestingly, the Ezh2 inhibitor has been used as an anti-cancer drug (Tazemetostat) to promote the tumoricidal immune response and is currently approved for the treatment of follicular lymphoma and epithelioid sarcoma [39]. Likewise, the downregulated Ezh2 enhanced the severity of a colitis mouse model through the facilitated nuclear factor kappa B (NF-κB) and tumor necrosis factor-alpha (TNF-α) [40]. However, there are some reports on reduced inflammatory responses by Ezh2 blockage through the upregulation of Suppressor of cytokine signaling 3 (*Socs3*) and the inhibition of Janus kinase/signal transducer and activator of transcription (JAK/STAT) pathway [41] which has been used to reduce the severity of atherosclerosis [42]. Hence, the impacts of *Ezh2* on inflammation and sepsis are still inconclusive. Moreover, *Ezh2* was one of the upregulated genes in macrophages with LPS tolerance found in our previous publication [43]. The screening of epigenetic inhibitors also demonstrated that the *Ezh2* inhibitor enhances TNF-α expression in the LPS-tolerant macrophages [44]. Thus, the control of macrophage responses through the manipulations of epigenetics is interesting for controlling sepsis immune responses [13,45].

Here, the influence of *Ezh2* on LPS was explored in both a single LPS activation (hyper-inflammatory responses) and after two stimulations (LPS tolerance), in vitro and in vivo, using conditional *Ezh2* deletion mice with a LysM-Cre system, which selectively affected *Ezh2* only in macrophages.

## 2. Results

### 2.1. An Influence of Epigenetic Alteration in LPS-Activated Macrophages, the Transcriptomic Analysis, and siRNA Experiments

A correlation between the epigenetic alteration and macrophage responses against LPS was evaluated through transcriptomic analysis. The differences between the bone-marrow-derived macrophages of wild-type (WT) mice with a single LPS stimulation and media control were demonstrated by the heat map graphic pattern (Figure 1A). The analysis of differentially expressed genes (DEG) indicated some genes with significant differences or a tendency of difference in epigenetic alteration between LPS-activated macrophages and the control group. First, there was a higher lysine deacetylase (epigenetic eraser) in LPS-stimulated macrophages, including *Hdac1* (histone deacetylase), *Kdm3a* (lysine demethylase 3A), and *Kdm6b* (lysine demethylase 6B), with a tendency of the upregulated *Sirt6* (sirtuin 6) (Figure 1B). Second, the lysine methyltransferase (epigenetic writer) was higher in LPS-stimulated cells with a prominent upregulated *Kmt5a* (lysine methyltransferase 5A) and an upregulated trend for *Ezh1* (histone-lysine N-methyltransferase-1) and *Ezh2* (histone-lysine N-methyltransferase-2) (Figure 1C). Third, there was downregulated serine-threonine/tyrosine protein kinase (epigenetic writer) after LPS stimulation with the decreased *Aurkb* (aurora kinase B) and a tendency for downregulation of *Aurkc* (aurora kinase C) (Figure 1D). Fourth, there was a trend of enhanced lysine ubiquitin ligase (epigenetic writer) and *Rnf2* (ring finger protein 2) in LPS-activated cells (Figure 1E). These data supported a correlation between epigenetic alteration through these enzymes with the macrophage responses against LPS.

### 2.2. The Reduced TNF-a and an Elevation of TNF-α and IL-6 in Ezh2 Null Macrophages with a Single LPS Stimulation and LPS Tolerance, Respectively

Due to the frequent alteration at H3K27 in activated macrophages which possibly be correlated with Ezh2 and PCR2 [31], the associations between *Ezh2* with one and two LPS stimulations were initially explored through the silencing of *Ezh2* gene by siRNA (*Ezh2* siRNA) on the macrophage cell line (RAW264.7) (Figure 2A). Notably, the difference between cytokine responses between one and two LPS stimulations was determined 2 days after the incubation (Figure 2A) to control the duration of culture in both groups. There is a difference in the duration of culture in the comparison between cytokines after the first and the second LPS stimulations, which are 1- and 2-days incubation periods, respectively. In a single LPS stimulation (N/LPS), the silencing of *Ezh2*, an inhibitory gene against cytokine production, in macrophages did not significantly change supernatant cytokines, including TNF-α, IL-6, and IL-10, and macrophage polarization (Figure 2B–I). The M1 polarization-associated genes, including interleukin-1β (IL-1β) and inducible nitric oxide synthase (iNOS) (Figure 2B–D), and M2 polarization-associated markers, including resistin-like molecule-1 (Fizz-1), arginase-1 (Arg-1), and transforming growth factor-β (TGF-β) (Figure 2E–I), were evaluated. In LPS tolerance (LPS/LPS), there was a reduction in supernatant cytokines (TNF-α and IL-6 but not IL-10) in both *Ezh* siRNA macrophages and the control when compared with the N/LPS stimulation, which supported the main characteristics of LPS tolerance as previously described [24,46,47]. However, there were higher TNF-α and IL-6 and lower IL-10 levels in *Ezh2*-silencing macrophages with LPS tolerance when compared with LPS tolerance in the control cells (Figure 2B–D). These data support the impact of the removal of an inhibitory gene (*Ezh2*), with similar markers on macrophage polarization (Figure 2E–I). Notably, the impact of LPS on cytokine production disappeared 48 h after the incubation, as all supernatant cytokines (TNF-α, IL-6, and IL-10) in macrophages activated with LPS followed by media control (LPS/N) protocol were similar to the control level (Appendix A).

### 2.3. Characteristics of Macrophages from Ezh2 Control (Ezhfl/fl; LysM-Cre^−/−^) or Ezh2 Null (Ezhfl/fl; LysM-Crecre^cre/−^) Mice after a Single LPS Stimulation and LPS Tolerance

To investigate the impact of *Ezh2* on LPS stimulation, macrophages from mice with conditional *Ezh2* deletion using the LysM-Cre system were used (Figure 3A). Similar to the *Ezh2* silencing with siRNA, the severity of LPS tolerance of macrophages from *Ezh2* null mice (*Ezh*^fl/fl^; LysM-Cre^cre/−^) was lower than LPS tolerance in the litter mate control (*Ezh*^fl/fl^; LysM-Cre^−/−^) (*Ezh2* control) as indicated by the higher supernatant TNF-α and IL-6 (but not IL-10) in macrophages from *Ezh2* null mice compared with the cells from *Ezh2* control (Figure 3B–D). The characteristics of LPS tolerance, as indicated by a lower cytokine production in LPS tolerance (LPS/LPS) compared with N/LPS, in macrophages from *Ezh2* null mice were demonstrated by reduced pro-inflammatory cytokines (TNF-α and IL-6) in the supernatant, while LPS tolerance in the cells from *Ezh2* control was indicated only by the lower IL-6 level (Figure 3B,C). Surprisingly, in a single LPS stimulation (N/LPS), supernatant IL-10 (an anti-inflammatory cytokine) was the only elevated cytokine in macrophages from *Ezh2* null mice (Figure 3D), supporting a possible less severe LPS tolerance from an absence of an inhibitory *Ezh2* gene [48]. In contrast, the supernatant TNF-α in *Ezh2* null macrophages with N/LPS protocol was lower than the cells from *Ezh2* control (Figure 3B), perhaps due to other inhibitory factors against the cytokine production. However, supernatant IL-6 from N/LPS-stimulated *Ezh2* null macrophages was not different from N/LPS-activated *Ezh2* control macrophages (Figure 3C), implying a lower impact of *Ezh2* siRNA in the inhibition of IL-6 compared with the reduction in TNF-α (Figure 3B).

### 2.4. A Possible Impact of Inhibitory Socs3 and Cell Energy Status in Ezh2 Null Macrophages with a Single LPS Stimulation and LPS Tolerance

Among several genes, *Ezh1*, *Ezh2*, and suppressor of cytokine signaling 3 (*Socs3*) were explored due to the possible inhibitory effect on inflammatory responses [49,50]. As such, *Ezh1* was downregulated in both N/LPS and LPS tolerance (LPS/LPS) in *Ezh2* null and control macrophages (Figure 4A). In control macrophages (*Ezh2* control), both N/LPS and LPS/LPS similarly upregulated *Ezh2* and *Socs3* when compared with the stimulation with media control (N/N) (Figure 4B,C). Meanwhile, in *Ezh2* null macrophages, there was a higher *Socs3* upregulation after both N/LPS and LPS tolerance compared with the activations in *Ezh2* control cells (Figure 4B,C). Additionally, *Socs3* upregulation in *Ezh2* null macrophages with N/LPS was higher than the activation by LPS tolerance (Figure 4C), implying a possibly more potent inhibition by *Socs3* after a single LPS stimulation, perhaps to neutralize LPS-induced hyper-inflammation from an absence of the *Ezh2* gene. Indeed, the *Ezh2* gene also controls *Socs3* expression, as the presence of *Ezh2* inhibits *Socs3* expression, as previously mentioned [50].

Because of the association between the energy status in macrophages with several activities, especially inflammatory responses [23,51,52,53,54], the extracellular flux analysis between *Ezh2* null macrophages and control cells was also explored. However, there was no statistical difference in the cell energy state (glycolysis and mitochondrial activities) among all groups of the experiments (Figure 4D,E). Additionally, the baseline of cell energy status between *Ezh2* null macrophages and control cells was similar, with a tendency for lower mitochondrial activities after activation by both N/LPS and LPS tolerance (LPS/LPS) in both groups (Figure 4D). Thus, *Ezh2* seems to have less impact on the cell energy status, despite several impacts on cytokine production.

### 2.5. A Less Pro-Inflammatory Response to LPS in Ezh2 Null Mice over the Control Mice, a Possible New Strategy against Sepsis-Induced Hyper-Inflammation by Ezh2 Interference

Due to the impacts of *Ezh2*-manipulated macrophages toward activation by a single LPS stimulation (N/LPS) and LPS tolerance (LPS/LPS), we further explored this in *Ezh2* null (*Ezh*^fl/fl^; LysM-Cre^cre/−^) and *Ezh2* control (*Ezh*^fl/fl^; LysM-Cre^−/−^) mice (Figure 5A). Serum TNF-α and IL-6, but not IL-10, was lower in *Ezh2* null mice compared with the *Ezh2* control group (Figure 5B–D), supporting impacts of the lower supernatant cytokines in LPS-activated *Ezh2* null macrophages compared to the control cells (Figure 3B). In LPS tolerance, the characteristics of lower serum cytokines after the second dose of LPS were demonstrated by both TNF-α and IL-6 in the control mice, as demonstrated in the open circles and open square that represented a single LPS stimulation and LPS tolerance, respectively (Figure 5B,C). Meanwhile, the LPS tolerance-induced lower cytokine production was demonstrated only by decreased serum IL-6 (but not TNF-α) in the *Ezh2* null mice as indicated by the blue circle (a single LPS stimulation) and red square (LPS tolerance) (Figure 5B,C). These data implied a less severe LPS tolerance in the *Ezh2* null group than in the control mice. Notably, the characteristic of LPS tolerance could not be demonstrated by serum IL-10 in all mouse strains (Figure 5D). Thus, the absence of *Ezh2* in mice was beneficial for anti-inflammation after LPS stimulation and in LPS tolerance when compared with the control mice.

## 3. Discussion

### 3.1. Epigenetic Regulation in LPS Stimulation, an Interesting Strategy in Immune Response Manipulation for Sepsis

The presence of lipopolysaccharide (LPS), a microbial molecule from Gram-negative bacteria, in blood has been clinically demonstrated as endotoxemia in several conditions, including sepsis [55,56,57], partly through gut barrier damage [1,22,24,58]. The activation of macrophages by LPS is possibly important in sepsis because macrophages are the innate immune cells responsible for the recognition of foreign molecules [51,59]. Among several alterations in LPS-stimulated macrophages, the epigenetic modifications, especially DNA methylation at the cytosine-phosphate-guanine (CpG) sites and the methylation at the N-terminal tails of histones, are critical regulators of chromatin structure that determine gene expression for responses, differentiation, and proliferation [60]. In epigenetic-induced histone modification, several key enzymes control chromatin accessibility by regulating methyl and acetyl marks at the tail of histone H3 by either installing (writers) or removing (erasers) these marks at histone. Although gene repression by the presence of trimethylation of H3K27 (H3K27me3) at promoter regions is well known in cancer [61], data on the influence of H3K27me3 in sepsis are still scarce [62]. After LPS stimulation, seven lysine methyltransferases, enzymes for the manipulation of methyl groups at lysine on histone protein as writers (adding the proteins) and erasers (removing these chemical tags), were increased, while only a few enzymes of other epigenetic processes were elevated. Perhaps, the methylation in LPS-activated macrophages might be more common than other epigenetic alteration processes that use lysine methylation (H3K27Me) to safeguard for the overwhelming production of inflammatory cytokines (immune hyper-responsiveness) during sepsis [33]. Between the key repressive and activating methyl mark at H3K27 by writers (*Ezh1/Ezh2*) and erasers (*Kmt5a/Kmt5b*) [35,63], respectively, *Ezh1, Ezh2,* and *Kmt5a* are in the lists of our data, highlighting the importance of lysine methylation in macrophage response to LPS. Indeed, the increased expression of *Ezh2* and H3K27 is demonstrated in the circulation of patients with sepsis, which is correlated with increased mortality [64,65]. Although *Ezh2* showed only a trend to be upregulated after LPS stimulation here, the clinical availability of Ezh2 inhibitors in the treatment of cancer [66] makes *Ezh2* an interesting target with an easy clinical translation. Theoretically, the presence of *Ezh2* induces gene repression (anti-inflammation), and the blockage of *Ezh2* might enhance inflammation that is possibly beneficial for LPS tolerance but worsens LPS-induced hyper-inflammation. Notably, the reduced inflammation in LPS tolerance might be too low and is inadequate for the proper inflammation necessary for microbial control. The attenuation of LPS tolerance might reduce the secondary infection [67,68,69].

### 3.2. Impact of Ezh2 in Sepsis-Hyper Inflammation and Immune Exhaustion

Although Ezh2 blockage might theoretically enhance pro-inflammation, a few reports using Ezh2 inhibitors and the conditioning of *Ezh2*-deleted mice in pneumococcal sepsis demonstrate beneficial impacts as an anti-inflammatory molecule, partly through the upregulation of Suppressor of Cytokine Signaling 3 (*Socs3*) gene [64,70]. In *Ezh2*-deficient macrophages, the reduced suppressive H3K27Me3 marks at the (*Socs3*) transcriptional start site (and distal enhancer) result in upregulated *Socs3*, despite the possibly more effective transcription of cytokine-producing genes due to the loss of methylation blockage of DNA reading by H3K27Me (Figure 6). As such, the cytosolic Socs3 inhibits the TLR-induced MyD88–TRAF6–NF-κB signaling pathway, partly through the enhanced ubiquitination and proteasomal degradation that suppresses the NF-κB–dependent inflammatory genes [35,49]. Likewise, Ezh2 inhibitors have also attenuated sepsis-induced intestinal disorders, multiple sclerosis, and glucose-activated peritoneal fibrosis in previous reports [50,62,71]. In contrast, the suppression of *Ezh2* worsens inflammatory bowel diseases and sepsis-induced muscle cell apoptosis [40,72,73,74], perhaps through the more effective pro-inflammatory cytokine production after the loss of DNA reading blockage by methylation (H3K27Me). Nevertheless, it is possible that H3K27Me is responsible for the control of some groups of cytokines more than other groups. For example, we demonstrated that LPS-activated *Ezh2* null macrophage induced higher IL-10, and lower TNF-a compared with the LPS-stimulated WT cells (Figure 3C). Hence, the impacts of *Ezh2* in sepsis are still inconclusive, and an exploration of the influence of *Ezh2* with LPS tolerance has never been carried out. Here, we supported an enhanced *Socs3* expression with the absence of *Ezh2* in both a single LPS stimulation (LPS response; N/LPS) and double LPS activation (LPS tolerance; LPS/LPS) using macrophages from *Ezh2* null mice (*Ezh*^fl/fl^*;* LysM-Cre^cre/−^) versus the control mice (*Ezh^fl/fl^;* LysM-Cre^−/−^) (Figure 4C). The upregulated *Socs3* was more prominent in the LPS tolerance of *Ezh2* null macrophages than in the WT cells, possibly due to differences in the H3K27Me abundance between a single versus the repeat LPS stimulation (LPS tolerance). Notably, *Ezh2* silencing by siRNA in RAW264.7 cells did not demonstrate anti-inflammation after N/LPS, which was different from the N/LPS in *Ezh2* null macrophages, possibly due to the limitation of the gene silencing by siRNA with the non-complete silencing of the interested genes. In LPS tolerance, there was an increase in cytokines in both *Ezh2* silencing by siRNA and *Ezh2* null macrophages, highlighting an improved severity of LPS tolerance despite a possible non-complete *Ezh2* silencing of the siRNA process. With an absence of the *Ezh2* gene, there was an IL10 upregulation in single LPS response (N/LPS) but not in LPS tolerance (LPS/LPS), which might be correlated with the more prominent *Socs3* expression in the N/LPS group compared with LPS tolerance. Indeed, *Socs3* might be correlated with IL-10 because (i) *Ezh2* inhibits *Socs3* expression and blockage of Ezh2 upregulates *Socs3* [50], (ii) IL10 directly upregulates *Socs3* expression in LPS-stimulated macrophages [75], and (iii) there is more prominent *Socs3* upregulation together with supernatant IL-10 in N/LPS than LPS tolerance (LPS/LPS) of *Ezh2* null macrophages (Figure 3D and Figure 4C). Additionally, *Socs3* activity might require IL10 to inhibit the inflammatory responses, as indicated by severe impairment of Socs3 and IL10 in macrophages with single LPS responses [76,77], upregulated IL-10 in LPS tolerance macrophages [78,79,80,81], and altered IL-10 in LPS-stimulated macrophages with an Ezh2 inhibitor (EPZ-6438) [82]. Moreover, reduced pro-inflammatory cytokines (TNF-α and IL-6) (Figure 3B,C) with an escalation in Socs3 and IL-10 in *Ezh2* null N/LPS macrophages (Figure 3D and Figure 4C) also supported a linkage among *Ezh2*, Socs3, and inflammatory responses. However, Socs3 can be regulated by both anti-inflammatory and pro-inflammatory cytokines [83], which is possibly driven by specific cytokines [84]. More studies on this topic are needed.

Hence, upregulated Socs3 might be responsible for the anti-inflammatory direction of *Ezh2* null macrophages and mice after a single LPS stimulation. In LPS tolerance, the loss of inhibitory Ezh2 in *Ezh2* null macrophages resulted in an elevation of both cytokine-producing genes, such as *NF-κB*, and *Socs3*. However, *Socs3* upregulation in the tolerance macrophages was not as prominent as the single LPS-stimulated macrophages leading to a lower Sosc3-inhibitory effect on cytokine production and higher cytokines in *Ezh2* null macrophages than control cells after LPS tolerance. Perhaps the balance between the pro-inflammatory NF-κB-dependent pathways and the counteracting *Socs3*-STAT3 anti-inflammation after TLR-4 activation [85] is the natural control of hyper-inflammation (Figure 6), and the absence of *Ezh2* tips the balance of the signaling by inducing the higher effective *Socs3* than *NF-κB*, leading to the prominent Sosc3-inhibitory effect, resulting in an anti-inflammatory stage. Because (i) COX-2 promotor is a CpG island, which is commonly found in DNA methylation, (ii) LPS enhanced *COX-2* expression in both mRNA and protein levels [86,87], and (iii) Ezh2 inhibition suppresses COX2 via Sosc3/STAT3 in macrophages and microglial cells [88,89]; with the well-known Socs3 and STAT3 correlation [90], the absence of Ezh2 might convert the Socs3-STAT3 function from a pro- to an anti-inflammatory phase in LPS tolerance. In LPS-activated mice (N/LPS), the anti-inflammatory effect of *Ezh2* depletion only in macrophages by the Cre-LoxP system [91] was demonstrated through the reduced serum TNF-α and IL-6 with relatively high serum IL-10 compared with the control mice. These data support that serum cytokine, in response to an LPS injection, was mainly produced from macrophages [92]. Due to lower levels of pro-inflammatory cytokines after LPS injection, the severity of LPS tolerance, as indicated by the difference between the first and second dose of LPS, in *Ezh2* null mice was lower than in the control. While the lower serum TNF-α and IL-6 levels were very obvious in the control mice with LPS tolerance, the relatively high serum TNF-α in LPS tolerance-stimulated *Ezh2* null mice was demonstrated through the non-different serum TNF-α between the first and second dose of LPS, indicating a less severe LPS tolerance. Despite the elevated supernatant IL-10 in *Ezh2* null macrophages with LPS tolerance, serum IL-10 was not increased in *Ezh2* null LPS tolerance mice, possibly due to an impact of LPS tolerance in other cells as IL-10 could be produced by innate immune cells, adaptive immune cells, and organ parenchymal cells [93,94,95,96]. Interestingly, the depletion of *Ezh2* in macrophages not only protected the mice from too high pro-inflammatory septic shock but, on the other hand, also safeguarded the mice from prominent LPS tolerance (fewer pro-inflammatory cytokines) which possibly correlated with better control on the secondary infection through the prevention of the microbial spread by an appropriate inflammation [97].

### 3.3. Clinical Aspect and Future Experiments

Although the sepsis immune responses are theoretically crudely divided into hyper-inflammation and immune exhaustion, the identification of the tips of the balance between these responses by several sophisticated biomarkers might be the future of sepsis immune modulation. For example, high serum IL-6 and IL-1 might be biomarkers for sepsis hyper-inflammation [98,99], while downregulated HLA-DR and viral reactivation, such as cytomegalovirus, which is the common dormant virus in the human host, possibly indicate sepsis immune exhaustion [100,101]. However, the use of drugs that are beneficial in both sepsis responses (hyper- and anti-inflammation) will be more convenient. From our data on mice with conditional *Ezh2* deletion, the blockage of Ezh2 might be one of the interesting drugs that are beneficial in both hyper-inflammation and immune exhaustion; however, reduced immune responses by *Ezh2* inhibition possibly did not induce immune exhaustion. Hence, Ezh2 blockage can be conveniently administered in both sepsis immune responses and is also clinically available [66]. The extended indication of Ezh2 blockage in sepsis, in addition to cancer treatment, might be beneficial. Nevertheless, LPS tolerance is only a subset of the sepsis-induced immune exhaustion [20] and the evaluation of Ezh2 blockage in sepsis is still scarce. Further experiments on the influence of Ezh2 deletion and Ezh2 inhibitors in sepsis are warranted.

Finally, there were several limitations in the current study. First, the results are from a limited number of mice in a proof-of-concept study and additional research is required to reach a solid conclusion. Second, the different ages and genders of the mice may affect the results and the experiments using female mice and/or different ages might result in a different conclusion. Third, additional information on the mechanisms behind the impact of Ezh2 is required to comprehend the influence on sepsis and LPS activation. Despite these various limitations, our data support the influence of Ezh2 in sepsis.

## 4. Materials and Methods

### 4.1. Macrophage Cell Line and Small Interfering RNA (siRNA)

Murine macrophage-like cells (RAW264.7; TIB-71), purchased from the American Type Culture Collection (ATCC, Manassas, VA, USA), were maintained in Dulbecco’s Modified Eagle’s Medium (DMEM; Cytiva HyClone) supplemented with 10% Fetal Bovine Serum (FBS) in a humidified incubator at 37 °C with 5% CO_2_. To initially explore an importance of *Ezh2* in responses against LPS, the small interfering RNA (siRNA) on *Ezh2* was evaluated. Briefly, RAW264.7 at 10^6^ cells/mL was seeded into 6-well plates and incubated overnight at 37 °C with 5% CO_2_. Then, the siRNA for *Ezh2* (DharmaconTM AccellTM, Horizon Discovery, Watwebeach, England, UK) was prepared with siRNA buffer before adding into the cells after removal of the cell culture media (the final concentration at 1 μM siRNA per well) at 37 °C with 5% CO_2_ for 48 h. The non-targeting pool siRNA (DharmaconTM AccellTM) was used as a control. Then, macrophages with Ezh2-siRNA and non-siRNA were activated by 3 different protocols, including (i) a single LPS stimulation, beginning with DMEM followed by LPS (100 ng/mL) 24 h later (N/LPS), or (ii) LPS tolerance, using the double stimulations by 100 ng/mL of LPS (LPS/LPS), or control (N/N) using the double DMEM incubation, before the sample collection (supernatant and cells). Notably, the supernatant of the stimulated cells in all groups was gently removed, and washed with DMEM, before the re-administration of LPS or DMEM as mentioned in previous publications [24,102,103]. Supernatant cytokines (TNF-α, IL-6, and IL-10) were evaluated by ELISA (Invitrogen, Carlsbad, CA, USA) and the gene expression was evaluated by quantitative real-time polymerase chain reaction (PCR) as previously described [104,105,106,107]. Briefly, the RNA was extracted from the cells with TRIzol Reagent (Invitrogen, Carlsbad, CA, USA) together with RNeasy Mini Kit (Qiagen, Hilden, Germany) as 1 mg of total RNA was used for cDNA synthesis with iScript reverse transcription supermix (Bio-Rad, Hercules, CA, USA). Quantitative real-time PCR was performed on a QuantStudio 5 real-time PCR system (Thermo Fisher Scientific, Waltham, MA, USA) using SsoAdvanced Universal SYBR Green Supermix (Bio-Rad, Hercules, CA, USA). Expression values were normalized to Beta-actin (*β-actin*) as an endogenous housekeeping gene and the fold change was calculated by the ∆∆Ct method. The primers used in this study are listed in Table 1.

### 4.2. Animal and Animal Model

The approved protocol (No. 017/2562) by the Institutional Animal Care and Use Committee of the Faculty of Medicine, Chulalongkorn University, Bangkok, Thailand according to the National Institutes of Health (NIH) criteria were used. Here, 8-week-old male wild-type (WT) C57BL/6 mice were purchased from Nomura Siam, Pathumwan, Bangkok, Thailand. In parallel, *Ezh2*^flox/flox^ and LyM-Cre^Cre/Cre^ mice were obtained from RIKEN BRC Experimental Animal Division (Ibaraki, Japan) and cross-bred until we reached an *Ezh2* littermate control (*Ezh*^fl/fl^; LysM-Cre^−/−^) or *Ezh2* null (*Ezh*^fl/fl^; LysM-Cre^cre/−^) in the F3 generation of the breeding protocol. The *Ezh2*^flox/flox^ mice have loxP sites upstream and downstream of the 2.7 kb SET domain, and were bred with LysM-Cre^Cre/Cre^ mice, the mice with a cre recombinase under the control of lysozyme M to target *Ezh2* for deletion in myeloid cells (macrophages and neutrophils). The offspring were either *Ezh2*^flox/flox^ with no LysM-Cre (*Ezh*^fl/fl^; LysM-Cre^−/−^), referred to as “the littermate controls or *Ezh2* control”, or were positive for the Cre driver (*Ezh2* null or Ezh^fl/fl^; LysM-Cre^cre/−^). For LPS activation, the conditional targeted Cre-positive mice (*Ezh2* null) were gender- and age-matched with floxed/floxed male littermate controls (*Ezh2* control) aged 8–10 weeks old. To genotype these mice on the loxP sites insertion, the following primers were used for *Ezh2*: reverse 1: 3′ of loxp: 5′-AGG GCA TCA GCC TGG CTGTA-3′; Forward 2: 5′ of loxp: 5′-TTA TTC ATA GAG CCA CCTGG-3′; Forward 3: left loxp: 5-ACG AAA CAG CTC CAG ATTCAG GG-3′ according to a previous publication [70]. The mice homozygous for the flox were selected and genotyped for the expression of LysM-Cre using the primers; Forward: 5′-CTTGGGCTGCCAGAATTCTC-3′; Reverse: 5′CCCAGAAATGCCA GATTACG-3′. Then, the mice were divided into 3 groups, including (i) LPS tolerance (LPS/LPS), intraperitoneal injection of 0.8 mg/kg LPS (Escherichia coli 026:B6) (Sigma-Aldrich, St. Louis, MO, USA) with another dose of 4 mg/kg LPS 48 h later, (ii) a single LPS stimulation, intraperitoneal phosphate buffer solution (PBS) injection followed by LPS (4 mg/kg) 48 h later, and (iii) control (N/N), double intraperitoneal PBS injection with 48 h duration between the dose. Notably, there was a lower dose of the first LPS administration (0.8 mg/kg) compared with the second dose (4 mg/kg) in the LPS tolerance protocol because the higher LPS in the first administration might result in the sustained elevated serum cytokines at 24 h of the first dose (before the 2nd dose LPS) which might interfere with the interpretation. After these protocols, blood was collected through (i) tail vein nicking at 1 and 3 h after the last injection and (ii) cardiac puncture under isoflurane anesthesia at 6 h of the protocol. Then, serum cytokines were evaluated by ELISA (Invitrogen, Carlsbad, CA, USA).

### 4.3. Bone-Marrow-Derived Macrophages and the Transcriptome Analysis

The RNA sequencing analysis was performed to determine the influence of epigenetic alteration in LPS-activated macrophages. As such, bone-marrow-derived macrophages were prepared from the femurs of wild-type (WT) mice using supplemented Dulbecco’s Modified Eagle’s Medium (DMEM) with a 20% conditioned medium of the L929 cells (ATCC CCL-1), which are fibroblasts used as a source of macrophage colony-stimulating factor as previously published [51,52,54,107]. Then, the macrophages at 5 × 10^4^ cells/well in supplemented DMEM (Thermo Fisher Scientific) were incubated in 5% carbon dioxide (CO_2_) at 37 °C for 24 h before being treated by LPS stimulation (100 mg/mL), and control DMEM for 24 h. For transcriptome analysis, the RNA from macrophages was extracted by RNeasy mini kit (Qiagen) and processing with the RNA sequencing of BGISEQ-50 platform based on triplicate macrophage samples as previously published [108]. The sequencing quality was determined using FastQC, and the raw sequencing reads were mapped and aligned against Mus musculus reference genome GRCm39 using STAR [109], followed by gene quantification against the reference mouse transcriptome using Kallisto [110]. Read counts were normalized and analyzed for differentially expressed genes (DEGs) using edgeR package [111] and limma-voom package [112,113]. Genes were considered significant expressions (*p*-value < 0.05) when the log2 value of fold change of expression was less than −2 or greater than 2, indicating down- or upregulation, respectively. Clustering of DEGs was performed based on Euclidean distance and the Ward.D2 method using the ComplexHeatmap package version 2.12.1 [114]. Expression levels in log2 (TPM; transcript count per million) of selected epigenetic-related genes [115] were compared between untreated and LPS-treated groups to determine statistical significance using the Wilcoxon test in ggpubr package [116], where a *p*-value less than 0.05 indicates statistical significance.

### 4.4. Bone-Marrow-Derived Macrophages and Extracellular Flux Analysis

Bone-marrow-derived macrophages from the *Ezh2* control (*Ezh*^fl/fl^; LysM-Cre^−/−^) or *Ezh2* null (*Ezh*^fl/fl^; LysM-Cre^cre/−^) mice were extracted from femurs before activation by N/LPS (single LPS stimulation), LPS/LPS (LPS tolerance), or N/N (control), similar to the protocol for siRNA macrophages, and we measured supernatant cytokines (TNF-α, IL-6, and IL-10) and gene expression for (*Ezh1, Ezh2,* and *Socs3*) by PCR as mentioned above. Additionally, the seahorse XFp Analyzers (Agilent, Santa Clara, CA, USA) were used to determine the cell energy status (extracellular flux analysis), with oxygen consumption rate (OCR) and extracellular acidification rate (ECAR) representing mitochondrial function (respiration) and glycolysis activity, respectively, following previous publications [23,53,108,117,118]. Briefly, the macrophages (1 × 10^5^ cells/well) at 24 h after the stimulations (N/N, N/LPS, and LPS/LPS) were incubated in Seahorse media (DMEM complemented with glucose, pyruvate, and L-glutamine) (Agilent, 103575–100) before activation by different metabolic interference compounds such as oligomycin, carbonyl cyanide-4-(trifluoromethoxy)-phenylhydrazone (FCCP), and rotenone/antimycin A for OCR evaluation. In parallel, glucose, oligomycin, and 2-Deoxy-d-glucose (2-DG) were used for ECAR measurement. The graphs of OCR and ECAR were demonstrated.

### 4.5. Statistical Analysis

The results are shown in mean ± S.E.M. All data were analyzed with GraphPad Prism version 6. Student’s *t*-test or one-way analysis of variance (ANOVA) with Tukey’s comparison test was used for the analysis of experiments with two and more than two groups, respectively. For all data sets, a *p*-value less than 0.05 was considered significant.

## 5. Conclusions

The transcriptomic analysis of LPS activation in wild-type macrophages demonstrated the correlation between several enzymes of epigenetic processes and the responses to sepsis. The reduced supernatant pro-inflammatory cytokines with a single LPS stimulation and the less severe LPS tolerance, as indicated by the lower differences in supernatant cytokines after the first and second dose of LPS, using a double LPS stimulation in *Ezh2* null (Ezh*^fl/fl^; LysM-Cre^cre/^*^−^) macrophages were compared with the control cell (Ezh*^fl/fl^; LysM-Cre*^−/−^). These data supported the possible benefits of Ezh2 blockage in both acute responses against LPS stimulation and LPS tolerance after several LPS activations in macrophages. Likewise, the less severe hyper-inflammation and LPS tolerance after a single and double LPS injection, respectively, in *Ezh2* null mice over the control mice also support the benefits of interference of Ezh2 during sepsis. More studies in other animal models and/or in patients with sepsis using clinically available drugs are encouraged.

## Figures and Tables

**Figure 1 ijms-24-05363-f001:**
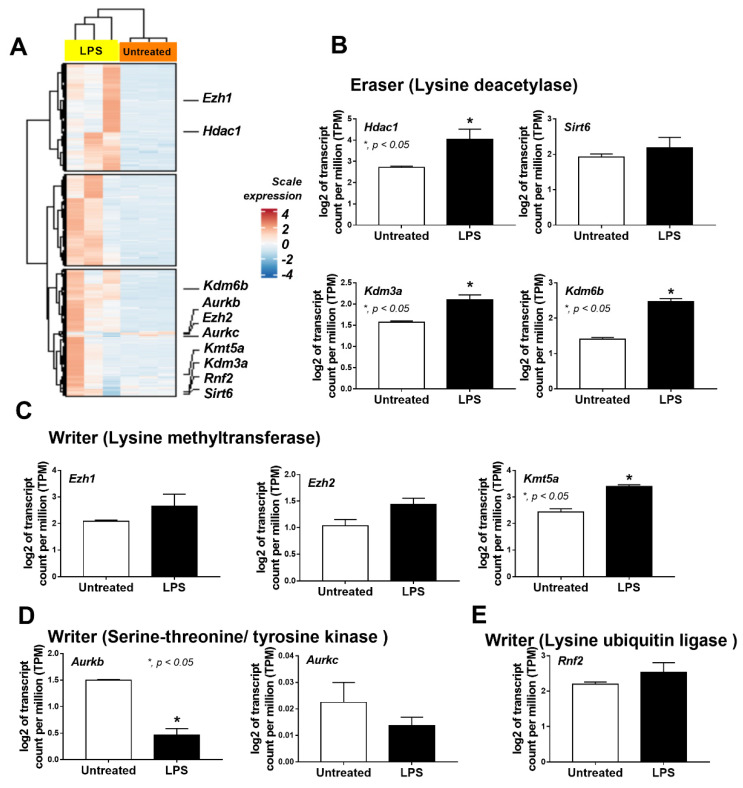
The transcriptome profiles and the log2 of the transcript count per million (TPM) of genes in bone-marrow-derived macrophages from wild-type mice 24 h after incubation by lipopolysaccharide (LPS) or media control (untreated) as indicated by the heatmap (**A**) and the bar plot showing some epigenetic-associated genes in several groups with statistically significant difference (or with a tendency of the difference), including (i) lysine deacetylase (eraser); *Hdac1* (histone deacetylase 1), *Sirt6* (sirtuin 6), *Kdm3a* (lysine demethylase 3A), *Kdm6b* (lysine demethylase 6b) (**B**), (ii) lysine methyltransferase (writer); *Ezh1* (histone-lysine N-methyltransferase-1), *Ezh2* (histone-lysine N-methyltransferase-2), and *Kmt5a* (lysine methyltransferase 5A) (**C**), (iii) serine-threonine/tyrosine kinase (writer); *Aurkb* (aurora kinase B) and *Aurkc* (aurora kinase C) (**D**), and (iv) lysine ubiquitin ligase (writer); *Rnf2* (ring finger protein 2) (**E**), are demonstrated. Macrophages were isolated from 3 different mice for the triplicate analysis. Mean ± SEM is presented with Student’s *t*-test analysis (*, *p* < 0.05).

**Figure 2 ijms-24-05363-f002:**
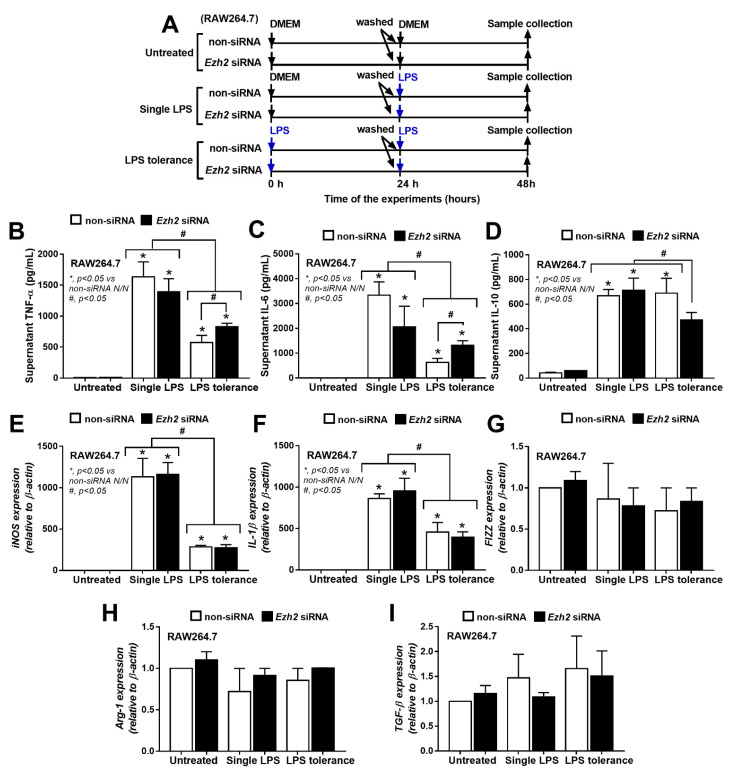
The schema of the experiments in a murine macrophage cell line (RAW264.7) with the silencing of *Ezh2* gene using small interfering RNA (*Ezh2* siRNA) or the control siRNA (non-targeting pool siRNA; non-siRNA) and activated by lipopolysaccharide (LPS) in a single protocol (N/LPS) which began with the culture media followed by LPS 24 h later or LPS tolerance (LPS/LPS) by two LPS stimulations, or control (N/N) using the culture media incubation twice (**A**). The characteristics of these macrophages with different protocols as indicated by supernatant cytokines (TNF-α, IL-6, and IL-10) (**B**–**D**), expression of pro-inflammatory genes of M1 polarization (iNOS and IL-1β) (**E**,**F**), and anti-inflammatory genes of M2 polarization (Fizz-1, Arg-1, and TGF-β) (**G**–**I**) are demonstrated. Triplicated independent experiments were performed. Mean ± SEM is presented with the one-way ANOVA followed by Tukey’s analysis (*, *p* < 0.05 vs. non-siRNA N/N and #, *p* < 0.05).

**Figure 3 ijms-24-05363-f003:**
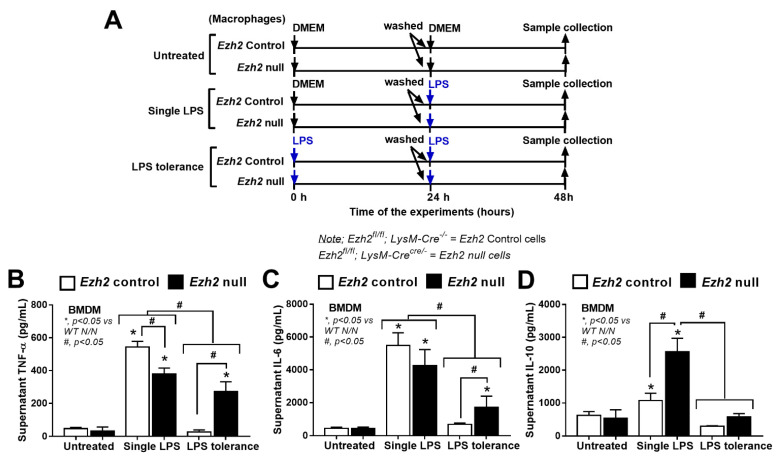
The schema of the experiments in bone-marrow-derived macrophages from *Ezh2* control (*Ezh*^fl/fl^; LysM-Cre^−/−^) or *Ezh2* null (*Ezh*^fl/fl^; LysM-Cre^cre/−^) mice after activation by lipopolysaccharide (LPS) in a single protocol (N/LPS), which began with the culture media followed by LPS 24 h later, or LPS tolerance (LPS/LPS) through two LPS stimulations, or control (N/N) using the culture media incubation twice (**A**). The characteristics of these macrophages with different protocols as indicated by supernatant cytokines (TNF-α, IL-6, and IL-10) (**B**–**D**) are also demonstrated. Triplicate independent experiments were performed. Mean ± SEM is presented with the one-way ANOVA followed by Tukey’s analysis (*, *p* < 0.05 vs. WT N/N and #, *p* < 0.05).

**Figure 4 ijms-24-05363-f004:**
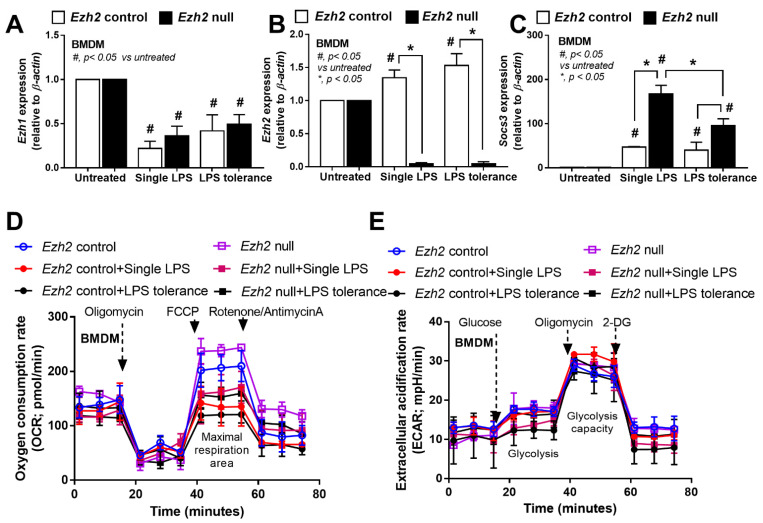
Characteristics of bone-marrow-derived macrophages (BMDM) from *Ezh2* control (*Ezh*^fl/fl^; LysM-Cre^−/−^) or *Ezh2* null (*Ezh*^fl/fl^; LysM-Cre ^cre/−^) mice 24 h after stimulation by lipopolysaccharide (LPS) tolerance (twice LPS stimulation; LPS/LPS) or a single LPS stimulation (N/LPS; started with phosphate buffer solution (PBS) followed by LPS) as indicated by the expression of several genes, including *Ezh1*, *Ezh2*, and *Socs3* (**A**–**C**) and the energy status of cells (extracellular flux analysis) (**D**,**E**). Independent triplicate experiments were performed. Mean ± SEM is presented with the one-way ANOVA followed by Tukey’s analysis (*, *p* < 0.05 and #, *p* < 0.05 vs. untreated).

**Figure 5 ijms-24-05363-f005:**
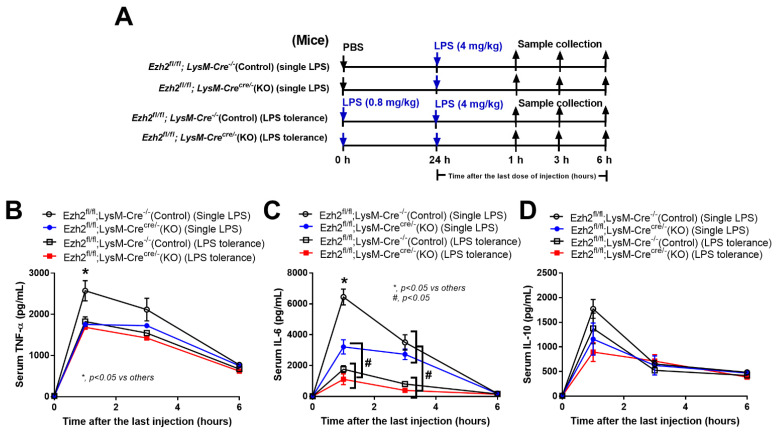
Schematic workflow (**A**) demonstrates the experimental groups, including lipopolysaccharide (LPS) tolerance; starting with LPS intraperitoneal (ip) injection (0.8 mg/kg) followed by LPS (4 mg/kg) (LPS tolerance; LPS/LPS), a single LPS stimulation; starting with phosphate buffer solution (PBS) followed by LPS (4 mg/kg) (a single LPS; N/LPS), in *Ezh2* control (*Ezh*^fl/fl^; LysM-Cre^−/−^) or *Ezh2* null (*Ezh*^fl/fl^; LysM-Cre^cre/−^) mice as indicated by serum cytokines (TNF-α, IL-6, and IL-10) (**B**–**D**) (*n* = 7/group or time-point). Mean ± SEM is presented with the one-way ANOVA followed by Tukey’s analysis (*, *p* < 0.05 vs. others and #, *p* < 0.05).

**Figure 6 ijms-24-05363-f006:**
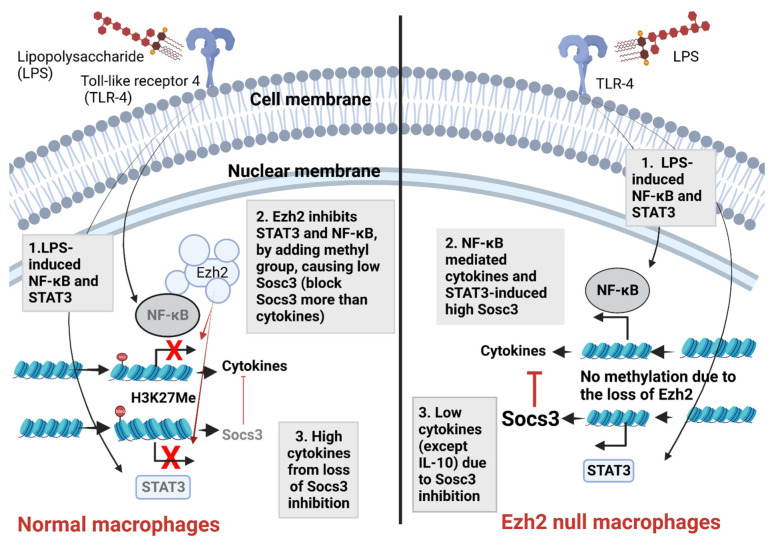
The proposed working hypothesis demonstrates the impact of Ezh2 (Enhancer of zeste homolog 2) in response against lipopolysaccharide (LPS) of macrophages. In normal macrophages (left side), LPS recognition by Toll-like receptor 4 (TLR-4) initiates several downstream signals, including nuclear factor-kappa B (NF-κB) and signal transducer and activator of transcription 3 (STAT3) [85] (number 1). The presence of Ezh2 causes the methylation on the 27th lysine of the histone (H3K27Me) resulting in the blockage of NF-κB and STAT3, which reduces cytokine production and suppressor of cytokine signaling 3 (Socs3; the inhibitor of NF-κB signaling [35,49]), respectively (number 2). The downregulated Sosc3 causes an increase in cytokines, highlighting the impact of the Ezh2 gene on the induction of LPS pro-inflammatory responses (number 3). Without Ezh2 (right side), both NF-κB and STAT3 effectively transcribe DNA into RNA of cytokines and Sosc3 (number 2); the upregulated Socs3 inhibits the synthesis of pro-inflammatory cytokines (TNF-α and IL-6), but not anti-inflammatory cytokines (IL-10) (number 3). More experiments are needed for a solid conclusion. Picture created by BioRender.com.

**Table 1 ijms-24-05363-t001:** Lists of primers used in the study.

Name	Forward	Reverse
Inducible nitric oxide synthase *(iNOS)*	5′-ACCCACATCTGGCAGAATGAG-3′	5′-AGCCATGACCTTTCGCATTAG-3′
Interleukin-1β *(IL-1β)*	5′-GAAATGCCACCTTTTGACAGTG-3′	5′-TGGATGCTCTCATCAGGACAG-3′
Tumor necrosis factor α *(TNF-α)*	5′-CCTCACACTCAGATCATCTTCTC-3′	5′-AGATCCATGCCGTTGGCCAG-3′
Arginase-1 *(Arg-1)*	5′-CTTGGCTTGCTTCGGAACTC-3′	5′-GGAGAAGGCGTTTGCTTAGTT-3′
Resistin-like molecule-α1 *(FIZZ-1)*	5′-GCCAGGTCCTGGAACCTTTC-3′	5′-GGAGCAGGGAGATGCAGATGA-3′
Transforming growth factor-β *(TGF-β)*	5′-CAGAGCTGCGCTTGCAGAG-3′	5′-GTCAGCAGCCGGTTACCAAG-3′
Enhancer of Zeste Homolog 1 (*Ezh1*)	5′-TGAAATCTGAGTATATGCGGC-3′	5′-AGATATCCTGGCTGTCGAAC-3′
Enhancer of Zeste Homolog 2 (*Ezh2*)	5′-GCCCACCTCGGAAATTTCCTGC-3′	5′-CAGAGCACCTGGGAGCTGCTG-3′
Suppressor of cytokine signaling 3 (*Socs3*)	5′-ATGGTCACCCACAGCAAGTT-3′	5′-AATCCGCTCTCCTGCAGCTT-3′
*β-actin*	5′-CGGTTCCGATGCCCTGAGGCTCTT-3′	5′-CGTCACACTTCATGATGGAATTGA-3′

## Data Availability

Not applicable.

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
