# Peer review of "The Regulatory Roles of Ezh2 in Response to Lipopolysaccharide (LPS) in Macrophages and Mice with Conditional Ezh2 Deletion with LysM-Cre System"

_ijms, 2023, doi:10.3390/ijms24065363_

Round 1

Author Response

your study contains some very interesting data. I have some minor points/questions that should be improved.

1.Please introduce all abbreviations used, some introductions of abbreviations such as NF-κB

and TNF are missing.

ANS: We thank the reviewer for the comment and correct it accordingly.

2.Figure 1A is unfortunately very blurred. Please revise this Figure.

ANS: We thank the reviewer for the comment and correct it accordingly.

3.Please indicate in the legends to the figures which statistical test you have used and clarify the

meaning of * and #.

ANS: We thank the reviewer for the comment and correct it accordingly. 

4.In the legend of figure 2 the reference to figure G-I is missing.

ANS: We thank the reviewer for the comment and correct it accordingly. 

5.How did you perform the stimulation? Which LPS concentration was used to treat the cells?

Did you remove the medium after 24 h, i.e. before the second stimulation, and add new

medium with the appropriate stimulant to the cells, or did you simply add LPS or PBS again?

Did you also determine the cytokine levels after the first stimulation? Do you see any

differences there? 

ANS: We thank the reviewer for the comment and add the information accordingly. Hence, we add more information in method (“Notably, supernatant of the stimulated cells in all groups were gently removed, washed with DMEM, before the re-administration of LPS or DMEM as mention in previous publications”) and in the graphic figures (Fig 2A and 3A). The cytokine responses after the 1st LPS in LPS/LPS was not determined but we determine cytokine after a single LPS stimulation through N/LPS group to match the duration of culture (48 h) to LPS/LPS group. Then, we mentioned this point in new result as following “Notably, the difference between cytokine responses between single and twice LPS stimulations was determined 2 days after the incubation (Fig 2A) to control the duration of culture in both groups. There is a difference in the duration of culture in the comparison between cytokine after the first and the second LPS stimulations which are 1- and 2-days incubation periods, respectively.”.

6.Wouldn't a control with LPS/N also be useful? This control would strengthen your results again. 

ANS: We thank the reviewer for the comment. The LPS/N between WT and Ezh2 deficient cells is indeed an interesting experiment to see if there is a difference in the decline of cytokines between both groups. However, there was no different in this aspect when compare with control indicating a rapid decline of cytokine production after stop LPS stimulation. Hence, we mention this information in the new version of manuscript in supplement fig and in result as following “Notably, the impact of LPS on cytokine production was disappear at 48 h after the incubation as all supernatant cytokines (TNF-α, IL-6, and IL-10) in macrophages activated with LPS followed by media control (LPS/N) protocol were similar to the control level (supplement figure 1A-C).”.

7.Why different LPS concentrations were used in group LPS/LPS mice? 

ANS: We thank the reviewer for the comment. The too high LPS injection might affect the second dose of LPS at 24 h later because we can not wash and remove LPS as done in the in vitro experiment. Then, we mention this aspect in the method section as following “Notably, a lower dose of the 1st LPS administration (0.8 mg/kg) compared with the 2nd dose (4 mg/kg) because the higher LPS in the 1st administration might result in the sustained elevated serum cytokines at 24 h of the 1st dose (before the 2nd dose LPS) which might interfere with the interpretation.”.  

8. Please explain why you used only male animals in your study? It would also be interesting to

know if female animals show the same effect. 

ANS: We thank the reviewer for the comment and put this remark as a limitation of our manuscript as the female mice might have a less severe sepsis due to the hormone effect. Hence, we put these sentences in the new discussion as following “the different mice's age and gender may affect the results and the experiments using female mice and/or different ages might result in a different conclusion”.

Reviewer 2 Report

The LPS tolerance has been studied for decades. The molecular mechanisms involve many aspects, including non-coding RNA, epigenetic modification etc. EZH2 also has been well documented, including regulating TLR4/MyD88/NF-kB pathway.  This manuscript used ezh2 knockout mice and macrophages to demonstrate the impact of EZH2 on the production of cytokine, TNF-a, IL-6 and IL-10 while LPS was administrated once or twice. The data are supportive to the conclusion.

However, the manuscript is not well-written and need major corrections

1.       The title shall be reorganised, such as:  The regulatory roles of EZH2 in response to LPS in vitro and in vivo.

2.        The ABSTRACT: remove the explanation in brackets and make them into proper sentences

3.       Main Text: the language styles and grammar need extensive correction, for example Line 102:” the impact of Ezh2 on LPS” is not a right expression. Many of these types of errors, so need a professional person to have a thorough check. LPS/LPS is a very confusing term, better to use different way to express it.

4.       Each figure for each session, not like 2.1  that has 2 figures

5.       Don’t box the panels in each figure

6.       In figure legends, the source of data, data types, i.g.  Mean±SD or median, and how many independent experiments as well as statistical analysis shall be clearly addressed. Explain * and# in each panel.

7.       The bar charts in Figure 2-4, shall be changed for easy reading. The x- must be readable. It is better to apply the style of figure 4A-C to all the bar chart and change the x- label.

8.       Panel A in figures 2, 3, and 4 are so hard to understand.

9.       Discussion: need to remove the subtitles. It seems too long and need to reorganise a bit. Need to discuss the limitation too.

10.   The diagram need simplify and more professional

Author Response

The LPS tolerance has been studied for decades. The molecular mechanisms involve many aspects, including non-coding RNA, epigenetic modification etc. EZH2 also has been well documented, including regulating TLR4/MyD88/NF-kB pathway.  This manuscript used ezh2 knockout mice and macrophages to demonstrate the impact of EZH2 on the production of cytokine, TNF-a, IL-6 and IL-10 while LPS was administrated once or twice. The data are supportive to the conclusion.

However, the manuscript is not well-written and need major corrections

1. The title shall be reorganised, such as: The regulatory roles of EZH2 in response to LPS in vitro and in vivo. 

ANS: We thank the reviewer for the comment and change the title into ”The regulatory roles of Ezh2 in response to lipopolysaccharide (LPS) in macrophages and mice with conditional Ezh2 deletion with LysM-Cre system”.

2. The ABSTRACT: remove the explanation in brackets and make them into proper sentences.

ANS: We thank the reviewer for the comment and correct it accordingly.

3. Main Text: the language styles and grammar need extensive correction, for example Line 102:” the impact of Ezh2 on LPS” is not a right expression. Many of these types of errors, so need a professional person to have a thorough check. LPS/LPS is a very confusing term, better to use different way to express it.

ANS: We thank the reviewer for the comment and send the manuscript to the English editing service of the university before submission as attached certification. We are willing to seek further help from other English editing services if the language still does not reach the minimum requirement of the journal. Also, we change "LPS/LPS" to "LPS tolerance" or mention both terms together in the new version manuscript. 

4. Each figure for each session, not like 2.1  that has 2 figures 

ANS: We thank the reviewer for the comment and correct it accordingly. The regulatory roles of Ezh2 in response to lipopolysaccharide (LPS) in macrophages and mice with conditional Ezh2 deletion with LysM-Cre system

5. Don’t box the panels in each figure 

ANS: We thank the reviewer for the comment and correct it accordingly.

6. In figure legends, the source of data, data types, i.g. Mean±SD or median, and how many independent experiments as well as statistical analysis shall be clearly addressed. Explain * and# in each panel. 

ANS: We thank the reviewer for the comment and correct it accordingly.

7. The bar charts in Figure 2-4, shall be changed for easy reading. The x- must be readable. It is better to apply the style of figure 4A-C to all the bar chart and change the x- label. 

ANS: We thank the reviewer for the comment and correct it accordingly.

8. Panel A in figures 2, 3, and 4 are so hard to understand. 

ANS: We thank the reviewer for the comment and revise it accordingly.

9. Discussion: need to remove the subtitles. It seems too long and need to reorganise a bit. Need to discuss the limitation too. 

ANS: We thank the reviewer for the comment add this part accordingly as following “Finally, there were several limitations in the current study. First, the results are from a limited number of mice in a proof-of-concept study requires additional research to get a solid conclusion. Second, the different mice's age and gender may affect the results and the experiments using female mice and/ or different ages might result in a different conclusion. Third, additional information on the mechanisms behind the im-pact of Ezh2 is required to comprehend the influence on sepsis and LPS activation. Despite these various limitations, our data support the influence of Ezh2 in sepsis.”.

10. The diagram need simplify and more professional. 

ANS: We thank the reviewer for the comment and revise it accordingly.

Round 2

Reviewer 1 Report

Dear authors,

thank you for the detailed answers to the questions and the revision of the manuscript.

Reviewer 2 Report

The authors have addressed all the comments and the manuscript has been greatly improved